# Innovative Cost-Effective Nano-NiCo_2_O_4_ Cathode Catalysts for Oxygen Reduction in Air–Cathode Microbial Electrochemical Systems

**DOI:** 10.3390/ijerph191811609

**Published:** 2022-09-15

**Authors:** Qixing Zhou, Ruixiang Li, Xiaolin Zhang, Tian Li

**Affiliations:** MOE Key Laboratory of Pollution Processes and Environmental Criteria/Tianjin Key Laboratory of Environmental Remediation and Pollution Control, College of Environmental Science and Engineering, Nankai University, Tianjin 300350, China

**Keywords:** advanced green material, nano-NiCo_2_O_4_, wastewater treatment, oxygen reduction, green energy conversion

## Abstract

Microbial electrochemical systems (MESs) can harvest bioelectricity from varieties of organic matter in wastewater through electroactive microorganisms. Oxygen reduction reaction (ORR) in a cathode plays an important role in guaranteeing high power generation, which can be enhanced by cathode catalysts. Herein, the tiny crystalline grain nanocrystal NiCo_2_O_4_ is prepared via the economic method and utilized as an effective catalyst in air–cathode MESs. The linear sweep voltammetry results indicate that the current density of 2% nano-NiCo_2_O_4_/AC cathode (5.05 A/m^2^) at 0 V increases by 20% compared to the control (4.21 A/m^2^). The cyclic voltammetries (CVs) and the electrochemical impedance spectroscopy (EIS) showed that the addition of nano-NiCo_2_O_4_ (2%) is efficient in boosting the redox activity. The polarization curves showed that the MESs with 2% nano-NiCo_2_O_4_/AC achieved the highest maximum power density (1661 ± 28 mW/m^2^), which was 1.11 and 1.22 times as much as that of AC and 5% nano-NiCo_2_O_4_. Moreover, the adulteration of nano-NiCo_2_O_4_ with a content of 2% can not only enable the electrical activity of the electrode to be more stable, but also reduce the cost for the same power generation in MESs. The synthetic nano-NiCo_2_O_4_ undoubtedly has great benefits for large-scale MESs in wastewater treatment.

## 1. Introduction

Microbial electrochemical systems (MESs) are one of the promising green energy conversion technologies that can convert the organic substrates (e.g., organic compounds, organic waste) in wastewater into electrical energy [1]. In MESs, electroactive microorganisms (EAMs) oxidize organic substrates to generate electrons and protons, then the electrons move to the cathode through an external circuit and combine with electron acceptors [2]. Meanwhile, the protons are transported through the membrane and react with oxygen to form a water molecule [3]. In recent years, air–cathode MESs have been widely used because this configuration can gain more electrical energy from oxygen in the air as the electron acceptor without aeration [4]. However, the performance of MESs is significantly limited by poor cathode oxygen reduction reaction (ORR) and high oxygen mass transfer resistance [5]. The electrode materials play an essential role in overcoming these problems [6,7]. Therefore, the research on materials of air–cathode becomes dominant in improving the performance of MESs. Generally, carbon materials such as carbon cloth [8], carbon nanofiber [9], and active carbon (AC) [10] are used as the cathode. Among these, AC is reported to have the advantage of a large surface area (1.2 × 10^5^ m^2^/m^2^ of projected surface area, 0.0026 USD/g) with higher porosity than graphite [11]. Unfortunately, the electrocatalytic activity of these cathodes made of pure carbonaceous materials is still poor when compared with other catalyst-loaded materials, and it cannot satisfy the requirement of the four-electron pathway for high power output [12]. Therefore, it is necessary to find a material which can be doped with an activated carbon electrode to generate the similar power of MESs at a similar or lower cost.

Platinum (Pt)-based catalysts are considered the most effective catalysts in MESs because of their high ORR kinetics, but their high cost hinders their practical applications [13]. Thus, it is urgent to find low-cost and durable catalysts with high performance to replace Pt. Some of the catalysts shown in Appendix A have been reported as efficient and active, including transition metal oxides [14], non-platinum metals [15], non-metallic elements [16], and conductive polymers [17]. Among these materials, transition metal compounds have attracted attention because of their excellent catalytic activity toward the ORR. Lv et al. used broccoli-like Co-Ni_2_P as an ORR catalyst with AC [18]. The maximum power density (MPD) with Co-Ni_2_P/AC reached 1814 mW/m^2^, which increased by 98% compared with the control. Ortiz-Martinez et al. used NiMn_2_O_4_ as catalysts, and the power output increased by 80% [19]. Compared with these transition metal oxides, NiCo_2_O_4_ possessed much better electrical conductivity and excellent ORR catalytic activity (Table 1). For example, Ge et al. constructed a NiCo_2_O_4_-modified AC air–cathode MFC. MPD of the MFC (1730 ± 14 mW/m^2^) was 2.28 times that of the bare AC cathode and was similar to the commercial Pt/C because of improvement to the ORR with reduced charger transfer resistance [20]. Although the addition of NiCo_2_O_4_ is effective in improving the power generation performance of MES, the amount of NiCo_2_O_4_ added in the process of fabricating electrodes has still not been thoroughly investigated, and it is unknown whether an excessive amount of NiCo_2_O_4_ will lead to a decrease in the economic efficiency. The cost of the cathode with 15 wt.% (45.62 mW/USD) NiCo_2_O_4_ is 1.2 times higher than that of the cathode with 10 wt.% NiCo_2_O_4_ (55.27 mW/USD). As the addition amount of NiCo_2_O4 increases, energy consumption and material costs may become negative for large-scale applications. Therefore, it is necessary to determine the ratio of NiCo_2_O_4_ to ensure MPD while keeping the cost as low as possible.

In this work, a method with the ultimate goal of economic efficiency was reported to fabricate nanocrystalline NiCo_2_O_4_ (<5 nm in diameter), and it was applied as a cathode catalyst in an air–cathode MES. Different concentrations (0, 2%, and 5%) of nano-NiCo_2_O_4_/AC composites were synthesized to optimize cathodes with the high electron transfer and ORR properties. The surface morphology and distribution of nano-NiCo_2_O_4_ catalysts were characterized with scanning electron microscopy (SEM) and X-ray diffraction (XRD). The catalytic performance of the synthesized cathodes was investigated by comprehensive electrochemical measurements in bioanode MESs. Finally, the production of the power density per dollar was evaluated.

## 2. Material and Methods

### 2.1. Synthesis of nano-NiCo_2_O_4_

NiCo_2_O_4_ was synthesized in three steps, as described [27]. First, 100 mg of CoCl_2_•6H_2_O and 61 mg of Ni(NO_3_)_2_•6H_2_O were dissolved in 20 mL of tertiary butanol, and the solution was added to 20 mL of tertiary butanol containing 250 mg of KOH and ultrasonically dispersed for 2 h. Secondly, the resulting solution was transferred into a 50 mL autoclave and heated at 95 °C for 3 h. After that, the autoclave was cooled to room temperature. Thirdly, the obtained precipitate was centrifuged and washed with water and ethanol. Then, the precipitate was dried at 50 °C overnight. Finally, the above dry product was ground with an agate mortar. Then, the precipitate was calcined in a tube furnace at 300 °C for 3 h under an argon atmosphere, and the final product, nano-NiCo_2_O_4_, was obtained.

### 2.2. Cathode Preparation

The air–cathode, consisting of a 0.5 mm catalysis layer (CL), an electron collector (stainless steel mesh, Type 304N, 60 meshes, Detiannuo Commercial Trade Co. Ltd., Tianjin, China), and a 0.5 mm gas diffusion layer (GDL), was prepared according to the rolling-press procedure reported by our group [28]. The CL was composed of AC (Xinsen Carbon Co. Ltd., Fujian, China) and polytetrafluoroethylene (PTFE) suspension (60 wt.%, Hesen, Shanghai, China) with an AC/PTFE ratio of 6. The GDL was made by rolling a mixture of carbon black (Jinqiushi Chemical Co. Ltd., Tianjin, China) and PTFE with a mass ratio of 3:7. For the nano-NiCo_2_O_4_-modified cathode, different mass ratios (0, 2 wt.%, 5 wt.%) of nano-NiCo_2_O_4_ were added during the stirring of AC and PTFE at 80 °C in a water bath in the preparation of CL. The cathodes were named control, 2% nano-NiCo_2_O_4_/AC_,_ and 5% nano-NiCo_2_O_4_/AC, respectively.

### 2.3. MES Construction and Operation

The experiments were performed on dual-chamber reactors (each cylinder chamber was 3 cm in diameter and 4 cm in length, with a net volume of 28 mL and cross-section area of 7 cm^2^) equipped with different NiCo_2_O_4_ proportioned air–cathodes [29]. All reactors were constructed by tightly bolting two cubes made of polymethyl methacrylate together. In addition, the chamber was separated by a cation exchange membrane (Ultrex CMI-7000, Membranes International Inc., Glen Rock, NJ, USA) which had been pretreated in electrolyte overnight before installment. Anodes were made of carbon fiber brushes (3 cm in diameter and 2 cm in length) pretreated with acetone and ethanol and then washed in deionized water [30].

All MESs were inoculated with the effluent from MESs operated for more than two years in our laboratory. The medium contained acetate (1.0 g/L), 50 mM phosphate buffer solution (PBS; NH_4_Cl 0.31 g/L, KCl 0.13 g/L, NaH_2_PO_4_ 2.13 g/L, Na_2_HPO_4_ 4.576 g/L), trace mineral (12.5 mL/L), and vitamin solution (5 mL/L) [31]. The medium was continuously bubbled with nitrogen/carbon dioxide gas (V:V, 4:1) for 20 min to remove dissolved oxygen before filling the reactors. All solutions were refreshed when cell voltages were less than 50 mV. Reactors were operated at 25 ± 1 °C in a constant temperature incubator in the air atmosphere, and all experiments were carried out in duplicate.

### 2.4. Sample Characterization and Electrochemical Measurements

The topography of the sample was examined with a scanning electron microscope (SEM, Shimadzu SS-550, Japan). X-ray diffraction (XRD, D/Max2000, Rigaku, Tokyo, Japan) was used to analyze the synthesized nano-NiCo_2_O_4_ powder.

The voltages (V) across the 1000 Ω external resistance were recorded automatically every minute, and the average voltage was collected every 30 min using a data acquisition system (PISO-813, ICP DAS Co., Ltd., China). Cyclic voltammetry (CV) of bioanodes was performed at the scan rate of 1 mV/s over the potential from 0.2 to −0.6 V (stabilization period of 300 s). Linear sweep voltammetry (LSV) was performed in 50 mM fresh PBS at the scan rate of 0.1 mV/s (stabilization period of 300 s) over the potential from 0.3 to −0.2 V vs. Ag/AgCl, as described previously [32]. Electrochemical impedance spectroscopy (EIS) was measured at open circuit potential (OCP) over a frequency range of 100 kHz to 0.1 Hz with a sinusoidal excitation signal of 10 mV. All CV, LSV, and EIS analyses were performed using a three-electrode system which was composed of a working electrode (WE), a counter electrode (CE, 1 cm^2^ pt), and a Ag/AgCl (4.0 M KCl) reference electrode (RE, +198 mV vs. the standard hydrogen electrode). All electrodes were purchased from Aida Hengsheng Technology Co., Ltd., Tianjin, China, and connected to a potentiostat (Autolab PGSTAT 302N, Metrohm, Herisau, Switzerland). The difference was that the WE of CV and EIS was the bioanode, while the WE of LSV was the cathode. Polarization curves were performed with the external resistance varying from 1000 to 50 Ω in MESs as previously described [33].

## 3. Results and Discussion

### 3.1. Characterization of NiCo_2_O_4_ Nanoparticles and Improvement to Cathodes

The XRD technique was used to determine the phase, crystalline, and purity of the samples. The crystal structure of nano-NiCo_2_O_4_ was observed with XRD measurements within the range of 20°–80°, as shown in Figure 1. The main diffraction peaks could be clearly observed at 2 Theta of 31.1°, 36.7°, 44.6°, 59.1°, 65.0°, and 77.3°, corresponding to the (220), (311), (400), (511), (440), and (533) peaks, respectively, which are similar to the XRD pattern of the cubic spinel NiCo_2_O_4_ (JCPDS Card: 20-0781) [34,35]. Additionally, no peaks of other compounds were detectable, indicating the pure crystalline grain nanocrystal NiCo_2_O_4_ was successfully formed.

SEM images were provided to investigate the surface morphology and the distribution of the synthesized catalysts on the electrode. Figure 2a,b show the SEM images of nanocrystal NiCo_2_O_4_ on the AC air–cathode. The NiCo_2_O_4_ nanoparticles were uniformly deposited on the surface of the cathode (Figure 2a). These nano-sized composites were reported as efficient in providing more electrochemical activity sites for oxygen access and accelerating the ORR of the electrode [36]. Figure 2c,d provide the SEM images of the AC air–cathode without NiCo_2_O_4_. In this cathode, AC provided more active sites for the redox reaction of NiCo_2_O_4_ by increasing the transport properties of ORR-relevant species [24].

The linear sweep voltammetry curves were utilized to evaluate the electrochemical performance and ORR activity of the AC air–cathode with different proportions of NiCo_2_O_4_ catalysts. As shown in Appendix A, the NiCo_2_O_4_/AC air–cathodes developed a higher current density than the control during the whole scan range. At a potential of 0 V, the current density produced by 5% NiCo_2_O_4_/AC (5.08 A/m^2^) was similar to that produced by 2% NiCo_2_O_4_/AC (5.05 A/m^2^), which showed an increase of 20% compared to the control (4.21 A/m^2^). These results indicated that the NiCo_2_O_4_ modification promoted the ORR of the cathode. In addition, the open circus potential (OCP) of air–cathodes was in the order of 5% NiCo_2_O_4_/AC (0.262 V) > 2% NiCo_2_O_4_/AC (0.249 V) > Control (0.225 V). The higher OCP indicated a higher electrochemical activity towards the ORR, indicating the effectiveness of the catalysts [23].

### 3.2. Power Generation Performance of MESs

The bare AC air–cathode, 2% NiCo_2_O_4_/AC air–cathode, and 5% NiCo2O_4_/AC air–cathode were used to assess the performance of power production in MESs. As presented in Figure 3, the formation of biofilm attached to the anode with the increase in voltage during 0–50 h. Although the three MESs with different cathodes did not appear to dramatically affect the time taken for power generation, the values of generated voltage are significantly different. The voltage of the MES with 2% NiCo_2_O_4_/AC air–cathode at 50 h was 0.32 V, which was two times that of the control. After 50 h, the biofilm gradually reached maturity with the different voltage outputs. Here, the maximum voltages of AC, 2% NiCo_2_O_4_/AC, and 5% NiCo_2_O_4_/AC were obtained in all MESs around 225 h with values of 0.57 V, 0.69 V, and 0.58 V, respectively. Compared with bare AC air–cathode MESs, the maximum voltage of the MES with 2% NiCo_2_O_4_ increased by 19%. The average voltages of AC, 2% NiCo_2_O_4_/AC, and 5% NiCo_2_O_4_/AC air–cathode MESs during 5 cycles reached 0.48 ± 0.02 V, 0.61 ± 0.04 V, and 0.40 ± 0.05 V, respectively. It could be observed that the maximum voltage and the average voltages values of AC air–cathode MESs were similar to 5% NiCo_2_O_4_/AC air–cathode MESs and significantly lower than 2% NiCo_2_O_4_/AC air–cathode MESs. Therefore, it was obvious that the proper introduction of transition metal oxides into AC cathodes greatly increased the maximum voltages. This phenomenon preliminarily confirmed that the NiCo_2_O_4_ as the cathode catalyst improved electrochemical performance and ORR activity to a certain extent compared to bare AC. Although NiCo_2_O_4_ had good redox activity, the conductivity of this synthesized material may be lower than AC. Thus, 5% NiCo_2_O_4_ in air–cathodes conversely affected the conductive performance and hindered the electron transfer process of MESs, thereby affecting the voltage output.

### 3.3. Electrochemical Characterization of MESs

The cyclic voltammetries (CVs) were performed to provide kinetic information for the heterogeneous electron-transfer reactions [37]. Here, CV analysis was carried out to investigate the performance of EAMs, which could be affected by the ORR activity of the electrodes. Sigmoidal catalytic waves and oxygen reduction peaks were observed in all CVs (Figure 4 and Appendix A), revealing the occurrence of redox processes on the anode. Although the CVs of all the MESs with different cathodes exhibited a similar reduction peak at −0.267 ± 0.009 V, the reduction current changed with the different NiCo_2_O_4_ proportions. The cathode with 2% NiCo_2_O_4_ had a higher electroactivity towards the ORR, which could improve the performance of the bioanode. The MES with a 2% NiCo_2_O_4_ cathode had the highest maximum current (~8.54 mA), which was 1.16 and 1.30 times that of 5% NiCo_2_O_4_ (~7.34 mA) and the control (~6.59 mA). Moreover, at the potential of 0 V, the currents of the MESs with 2% NiCo_2_O_4_/AC, 5% NiCo_2_O_4_/AC, and control were 5.42 mA, 3.61 mA, and 4.78 mA, respectively, meaning that the proper addition of NiCo_2_O_4_ was efficient to boost ORR activity and thus improve the performance of MESs [38]. The addition of NiCo_2_O_4_ increased the activity sites for oxygen and accelerated electron transfer at the interface of electrodes [26].

To better investigate the impact of the catalyst on the charge transport of cathodes, the electrochemical impedance spectroscopy (EIS) experiments were conducted after one hour with OCP. Nyquist plots and the corresponding equivalent circuits are shown in Figure 5. The EIS was almost similarly shaped among these three MESs, with a well-defined single semicircle over the high-frequency range. There were few changes in ohmic resistance (R_s_) with the value of 13.8 ± 0.2 Ω due to the use of the same reactor, fixed RE, and PBS solution in EIS tests. Differences in total resistance were mainly contributed by the charge transfer resistance (R_p_). R_p_ usually represents the resistance of electrochemical reactions on the electrode [39]. As reported, the smaller the R_p_, the faster the rate of electron transfer, which is beneficial to ORR activity [35]. R_p_ of 5% NiCo_2_O_4_/AC was 4.58 Ω, which was 1.7 and 1.8 times as much as that of the control (2.69 Ω), and 2% NiCo_2_O_4_/AC (2.55 Ω). It has been demonstrated that the smaller the charge transfer resistance (R_p_), the faster the rate of electron transfer from microorganisms to the anode, which is beneficial to ORR activity, and therefore results in the improvement of current densities [35]. Therefore, the MES with a high proportion of NiCo_2_O_4_ (5%) could not improve the ORR activity.

To further explore the electrochemical activity of the MES, the polarization curves were measured. The power density and the polarization curves for the MESs with different cathodic catalysts are illustrated in Figure 6. As shown in Figure 6a, the cathode with bare AC produced an MPD of 1499 ± 36 mW/m^2^, which was close to the former report [32,40], while the MES with 2% NiCo_2_O_4_/AC obtained the highest MPD (1661 ± 28 mW/m^2^), 1.11 times as much as that of bare AC. However, the MPD of MESs with 5% NiCo_2_O_4_/AC was 1364 ± 25 mW/m^2^. The high MPD was related to both anode microbial activity and cathode catalysts with the operation of MESs. The voltage values decreased in all MESs with the increase in the current density. Therefore, the internal resistance of the MESs could be estimated by calculating the slope of the curves. From the linear fitting results, the slopes of the different MESs in this study were almost the same (−23.50 ± 1.37) and showed a similar trend in EIS; this could be explained as the cathode with 2% NiCo_2_O_4_ enabled more active sites and conductive channels to electron transfer in contact with the electrolyte [26].

The individual polarization curve of cathodes and anodes during the operation is shown in Figure 6b. The cathode potentials of 2% NiCo_2_O_4_/AC were significantly higher than that of the control, indicating the 2% NiCo_2_O_4_/AC exhibited a better catalytic performance and ORR activities in MESs. For example, at a current density of 1.3 A/m^2^, the cathode potential of 2% NiCo_2_O_4_/AC (21.89 mV) was 6.10 times that of the control (3.59 mV). In addition, the anode potentials exhibited almost the same trend, which provided further evidence that the improved performance of MESs was mainly caused by the new cathode.

## 4. Cost–Benefit Analysis

The low cost plays a determining factor in the application of MESs from lab-scale to large-scale, though some mid-scale MESs are operated [41]. Therefore, the cost–benefit of an MES can be improved using low-cost materials that do not significantly sacrifice performance. Due to the low cost, natural abundance, better electrochemical activity, and higher conductivity compared with nickel and cobalt oxide, NiCo_2_O_4_ has been accepted as a catalyst for electrode modification. Here, the 2% nano-NiCo_2_O_4_/AC air–cathodes showed superior electrochemical activities and promising performance. Considering the cost of necessary chemicals and energy (October 2020), 2% nano-NiCo_2_O_4_ air–cathodes cost 31 USD/m^2^, while bare ACs cost 30 USD/m^2^ [42]. Although the total costs are similar, significant differences are found in power generated per unit cost. The cost of power generation of 2% nano-NiCo_2_O_4_ air–cathodes was 106.85 mW/USD, which was 1.11 times as much as that (96.31 mW/USD) of AC air–cathodes. These results indicate that a 2% nano-NiCo_2_O_4_/AC air–cathode could be the better alternative for large-scale applications.

## 5. Conclusions

The highly efficient nanocrystal NiCo_2_O_4_ (with a diameter lower than 5 μm) as superior ORR catalysts for air–cathode MESs were synthesized via the economic method. The MPD of the MES with the optimal modifying ratio (2%) of nano-NiCo_2_O_4_ was found to be 1.11 and 1.22 times as much as that of bare AC air–cathode MESs and 5% nano-NiCo_2_O_4_/AC MESs. The enhancement is mainly attributed to the promotion of electron transfer and reduction in activation energy for the ORR. However, the higher NiCo_2_O_4_ addition may hinder the availability of the terminal electron acceptor-oxygen and increase the cost. These results indicate that the new nano-NiCo_2_O_4_ catalyst is an affordable and easily prepared cathode catalyst. Meanwhile, the amount of nano-NiCo_2_O_4_ catalyst addition needs to be considered comprehensively in preparing electrodes, which is important to reduce the cost and improve the power generation of large-scale MESs.

## Figures and Tables

**Figure 1 ijerph-19-11609-f001:**
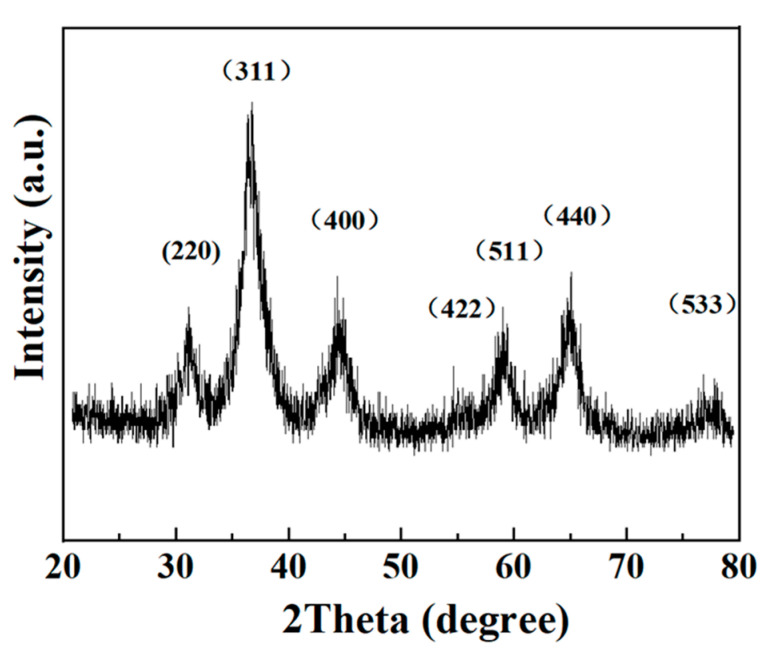
X-ray diffraction patterns of NiCo_2_O_4_.

**Figure 2 ijerph-19-11609-f002:**
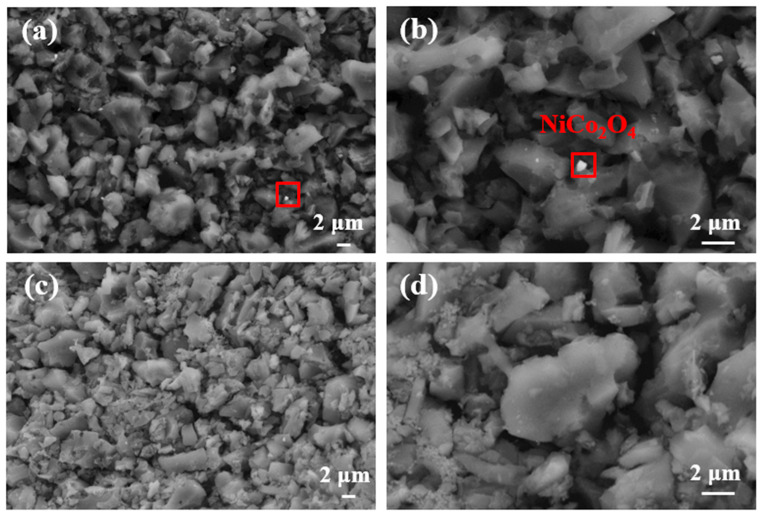
SEM images: (**a**) NiCo_2_O_4_ on the AC air–cathode; (**b**) magnified graph of the selected area in image a; (**c**) SEM image of bare AC air–cathode; and (**d**) image under the same multiple as image B of bare AC air–cathodes. The red square is considered to be the NiCo_2_O_4_.

**Figure 3 ijerph-19-11609-f003:**
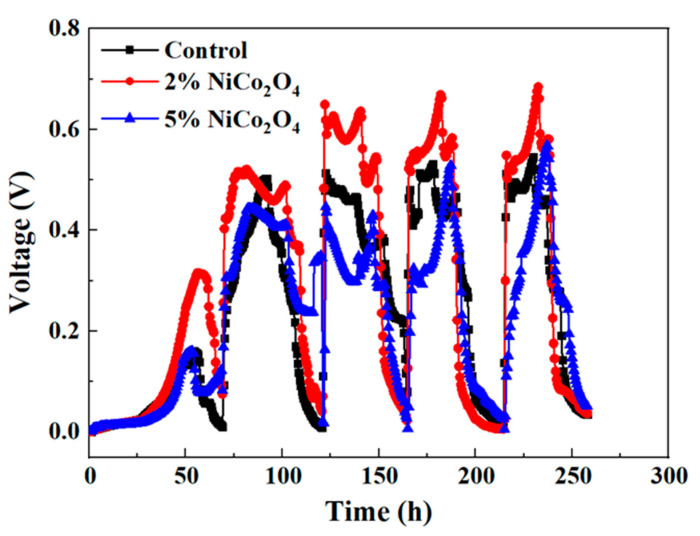
The output voltage of MESs with different cathodes.

**Figure 4 ijerph-19-11609-f004:**
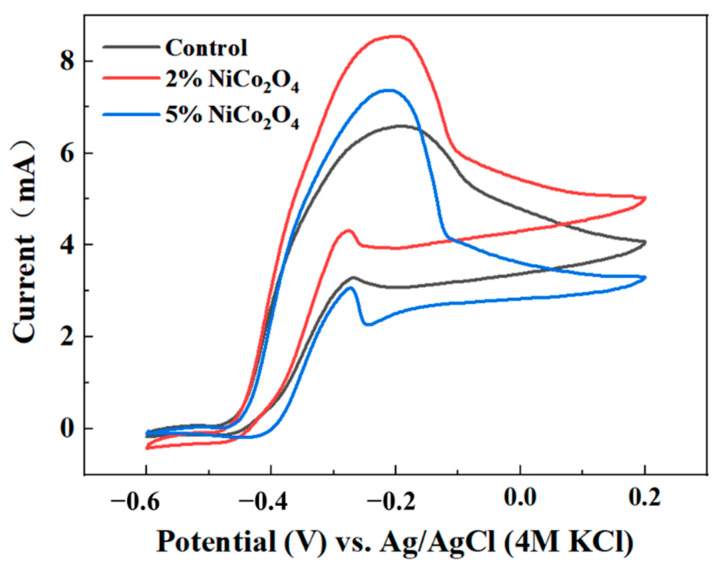
The cyclic voltammetry of the cathodes with 0, 2%, and 5% NiCo_2_O_4_ under 1 mV/s.

**Figure 5 ijerph-19-11609-f005:**
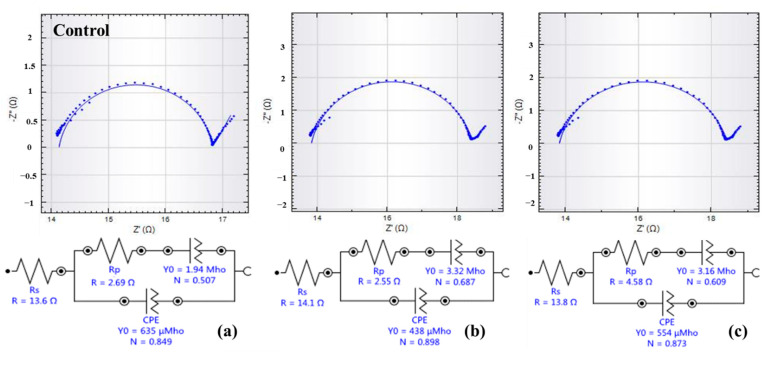
Nyquist plots and the equivalent circuit of the control (**a**), 2% NiCo_2_O_4_/AC (**b**), and 5% NiCo_2_O_4_/AC (**c**).

**Figure 6 ijerph-19-11609-f006:**
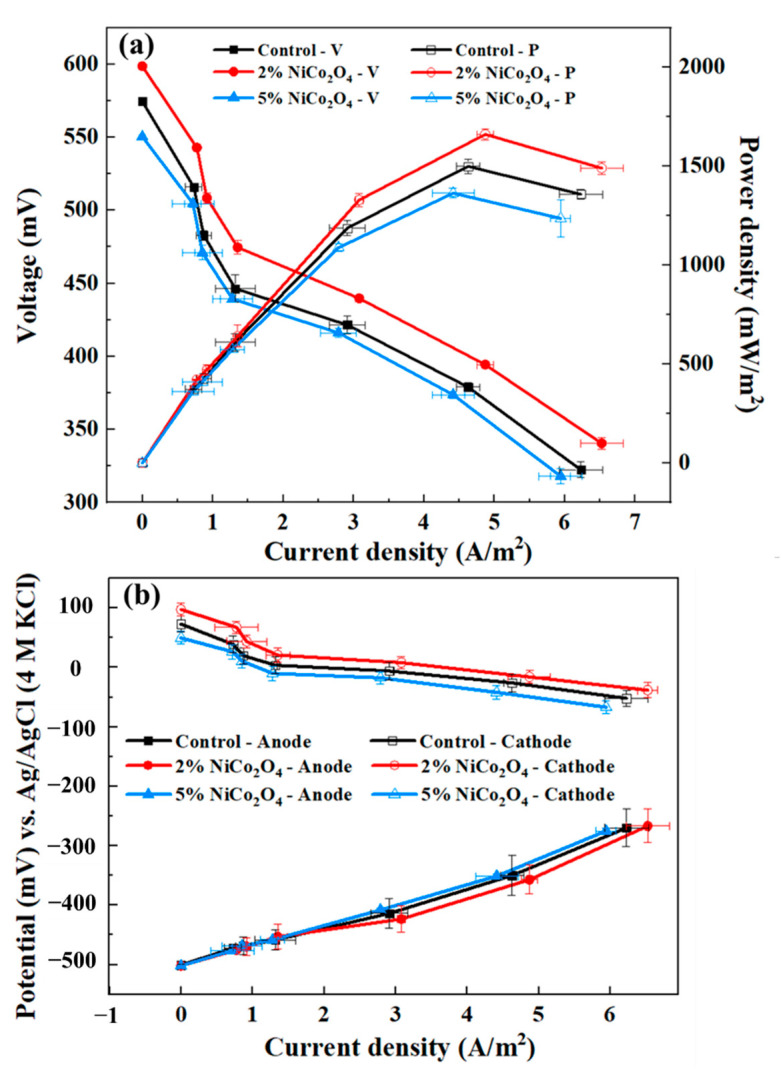
(**a**) Polarization and power density curves, and (**b**) individual polarization curves of cathodes and anodes during operation.

**Table 1 ijerph-19-11609-t001:** The published data of the NiCo_2_O_4_-modified cathode.

Configuration	Application	Cathode Materials	Synthesis Methods	Power Density (mW/m^2^)	Ref.
Single-chamber MFC	Power generation	Active carbon and nano urchin-like NiCo_2_O_4_	Rolling–press	1730	[20]
Dual-chamber MFC	Power generation	Graphite plate and NiCo_2_O_4_	Electrophoretic deposition	72	[21]
Dual-chamber MEC	Biohydrogen production	Nickel foam and NiCo_2_O_4_-graphene nanocomposites	Polymer binder	/	[22]
Single-chamber MFC	Power generation	Active carbon and Co_3_O_4_ and NiCo_2_O_4_	Rolling–press	1810	[23]
Single-chamber MFC	Power generation	Active carbon and NiCo_2_O_4_	Hydrothermal	1676	[24]
Single-chamber MFC	Power generation	Carbon cloth and NiCo_2_O_4_	Electrophoretic deposition	645	[25]
Dual-chamber MFC	Power generation	Acetylene black and NiCo2O4	Hydrothermal and coating	1250	[26]

## Data Availability

The datasets used and analysed during the current study are available from the corresponding author upon reasonable request.

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
