# Peer review of "Innovative Cost-Effective Nano-NiCo2O4 Cathode Catalysts for Oxygen Reduction in Air–Cathode Microbial Electrochemical Systems"

_ijerph, 2022, doi:10.3390/ijerph191811609_

Round 1

Reviewer 1 Report

I find the study useful but I have doubt on the Novelty of the work. There are lots of work done in the nano-NiCo2O4 modified cathode for to achieve excellent electrochemical performance in an air-cathode microbial electrochemical system. For example:

Khajeh, R. T., Aber, S., & Zarei, M. (2020). Comparison of NiCo2O4, CoNiAl-LDH, and CoNiAl-LDH@ NiCo2O4 performances as ORR catalysts in MFC cathode. Renewable Energy, 154, 1263-1271.

Li, M., Zhang, H., Xiao, T., Zhang, B., Yan, J., Chen, D., & Chen, Y. (2017). Rose flower-like nitrogen-doped NiCo2O4/carbon used as cathode electrocatalyst for oxygen reduction in air cathode microbial fuel cell. Electrochimica Acta, 258, 1219-1227

Cao, C., Wei, L., Wang, G., & Shen, J. (2017). In-situ growing NiCo2O4 nanoplatelets on carbon cloth as binder-free catalyst air-cathode for high-performance microbial fuel cells. Electrochimica Acta, 231, 609-616. Narayanasamy, S., & Jayaprakash, J. (2021).

Carbon cloth/nickel cobaltite (NiCo2O4)/polyaniline (PANI) composite electrodes: Preparation, characterization, and application in microbial fuel cells. Fuel, 301, 121016. 

How this work different from the work done before authors should add a section on it in introduction. 

I will suggest authors to provide a short review table on the use, synthesis, and efficiency of nano-NiCo2O4 modified cathode. although a brief review is presented in the supplementary table 3. But in this table, the catalyst used are all different, as there are lots study done with the NiCo2O4 modified cathode, I suggest to compare the efficiencies with the NiCo2O4 modified cathode also and provide a review column explaining how these methods are different and useful. And provide this Table in the main manuscript.

I will suggest to keep uniform font size and font style in each figure values, axis titles etc. For example-

Fig 1 and Fig 6: Font of the axis title and values are too large.

Fig 3 and Fig 6: Why axis values are bold?

Fig.5 Values in the figures are too small and quality of picture can be improved.

Cost benefit analysis: There is one line in this section "Although the cost was similar, the big difference was found in output per money. What does it mean? I will suggest author to give an comparative table on the cost of this process with the cost of other reported literature on the synthesis and use of NiCo2O4 in MCF.

The typos and grammar needs to be recheck throughout the manuscript.

Overall, I appreciate the authors for nice compilation of manuscript except few figure presentations. 

Author Response

We sincerely thank you for thoroughly examining our manuscript and providing very helpful comments to guide our revision. The revision has eliminated a number of errors and strengthened the overall manuscript, for which we are grateful.

The detailed responses to the reviewers’ comments are presented in the attached file. The line number of the location of each revision in the revised manuscript is described in each response and refers to the attached file entitled “Revised manuscript-marked”. 

We sincerely hope that this revised manuscript has addressed all your comments and suggestions. We appreciated for reviewers’ warm work earnestly, and hope that the correction will meet with approval. Once again, thank you very much for your comments and suggestions.

We would like to thank the referee again for taking the time to review our manuscript.

Reviewer 2 Report

The results presented are important and should be published. However, my concerns are listed below:

Line 86: Please underline the novelty and originality of the study.

Lines 217-223: The month and the year of when the costs were assessed should be pointed out in the text.

Line 232: What are next steps and how this could affect future studies and industry.

Good luck.

Reviewer 3 Report

Interesting project relating to the production of a novel nano-NiCo2O4 modified cathode in order to achieve good electrochemical performance in an air-cathode microbial electrochemical system (MES) by a cost-effective method.

However a few issues arise from the analysis of the paper:

Review the English language throughout the paper.

Avoid the use of the first person: “we”.

Rearrange the structure of the paper. Chapter 5 (Materials and Methods) should be Chapter 2.

The authors refer insistently that the method is cost-effective. However the cost-benefit analysis (chapter 3) seems somewhat simplistic. Develop further this analysis.

Refer the specific role of each author in the development of the paper.

Round 2

Reviewer 3 Report

In the revised manuscript, the authors address satisfactorily the recommendations received. I appreciate the effort, and now, I invite the authors to check their work carefully before resubmission, particularly, in taking special consideration to the English proofreading that has been carried out. I think that after this step the manuscript will be ready for acceptance.